# Economic Efficiency of the Implementation of Digital Technologies in Energy Power

**Victoria Galkovskaya [1,*] and Mariia Volos [2]**

1 Department of Economics, National Research Nuclear University, 249040 Obninsk, Russia
2 NPP Operation Organization Department, Rosatom Technical Academy, 249031 Obninsk, Russia
* Correspondence: summersvida@gmail.com; Tel.: +7-910-604-5752

**Abstract:** The present research prioritized the most important direction of energy power transformation—energy sector digitalization—and its contribution to the achievement of the sustainable development goals focused on climate change mitigation and responsible consumption and production. The authors evaluated the economic efficiency of the implementation of digital tools due to the decrease in energy production costs. The evaluation was performed using the energy cost calculation method before and after the implementation of digital technology, and the digital technology cost calculation method based on the technology readiness level, introduced for the first time. The study demonstrates the advantages of energy power digitalization worldwide, including the achievement of the declared goals of sustainable development and the transition to low-carbon energy. The scientific results obtained are valuable for scholars who carry out research in the area of the efficiency of economic digitalization, since they hold for all countries and types of energy generation. The directions of further research include the development of an analytical notation to show that the digital technology cost also depends on the speed and volume of its distribution, as well as the level of technological development of material and technical resources in the industry and in the country, and also the labor and capital availability.

**Keywords:** sustainable development goals; power industry digitalization; digital technologies; economic efficiency of digitalization; energy cost calculation; digital technology cost calculation method

## 1. Introduction

The digitalization of the Russian economy ensures the achievement of sustainable development goals [1] as well as realization of the low-carbon energy scenario [2,3], and is based on seven roadmaps designed for the development of cross-cutting digital technologies. Cross-cutting digital technologies simultaneously embrace several trends or industrial sectors and consist of quantum technologies; wireless communication technologies; virtual, augmented, and mixed reality (VR/AR/MR); neurotechnologies and Artificial Intelligence (including computer vision); distributed ledger technology (including smart contracts and Blockchain); new manufacturing technologies (including smart design and smart manufacturing), and robotics and sensorics [4–10].

The study of the scientific community's interest in end-to-end digital technologies in economic sectors based on the results of the analysis of scientometric databases is a separate field of scientific research. However, the following end-to-end technologies are of most interest in the field of power energy in Russia, as reflected in the results of scientific research: Big Data, neurotechnologies and Artificial Intelligence, new production technologies, industrial internet, robotics components and sensorics, wireless communication technologies, and virtual and augmented reality technologies. Meanwhile, distributed registry systems and quantum technologies lag significantly behind [11].

Nevertheless, neurotechnologies, Artificial Intelligence, and quantum technologies have limited practical application in the Russian electric power industry—the former due

to underdeveloped neural network information compression methods, as well as the lack of a regulatory framework, and the latter due to the low technology readiness level. Each digital technology has a different level of technological readiness, as shown in Appendix A, which affects the digital technology cost. However, the analytical justification of the impact of the readiness level on the digital technology cost is not investigated in the academic works devoted to economic digitalization, neither in Russia nor in the world as a whole. This provided the authors with the opportunity to fill this gap in the present study.

The rest of the paper is structured as follows: the literature review presents the outlook of the research relevant to the subject, materials and methods section describes the methodology followed by extensive results and discussion section, the last section summarizes.

*Literature Review*

In light of the escalating interest in the digitalization of the economy, a substantial number of scholars have explored different aspects of digital applications. Since the present study focuses on the energy sector, the literature review covers academic papers devoted to advances in certain digital technologies in the power industry. The substantial amount of work on the digitalization of the energy sector offers comprehensive reviews exploring the benefits and limitations of digital tools in the energy sector [12–18].

The most exhaustive overview of the main digital tools used in the energy sector is presented in [13,14,19,20]. The paper [19] gives an accurate description of the emerging digital concepts, such as Smart Grid, Smart Market, Blockchain, Artificial Intelligence, Cloud Computing, and Machine Learning, and discusses the potential risks of their application, with a focus on Germany. The review [13] gives insights into the Artificial Intelligence (AI), Machine Learning, Deep Learning, and Blockchain digital technologies and their applications in the power industry based on a theoretical review of 40 startups in the energy sector. In [14], the concept of smart machines in the energy sector is discussed and recommendations for their utilization considering specific requirements for implementation are given. In [20], an in-depth review of forty-seven scholars' work on the digitalization of the Russian energy system is presented. The paper determines the main directions of the current research and identifies the future research areas on the subject.

Research on the advantages of the Smart Grid is presented in [21,22] for the Pakistani and Danish energy markets, respectively. The necessities and challenges of connecting renewable sources of energy to the electricity grid utilizing the Smart Grid technology in the Danish energy market are addressed in [21]. A review of emerging techniques in the control of Smart Grids is presented for the Pakistani energy market [22]. The study reveals the possibility of using the Smart Grid as a tool to combat blackouts. The authors argue that the results are valid for other developing countries.

The opportunities and limitations of Blockchain technology are explored in [15,16,23,24]. The paper [15] offers a holistic overview of current and future applications of Blockchain technology in the electrical energy sector. The main contribution of the paper is the identification of the most prominent Blockchain technologies for utilization in power systems, as well as areas where Blockchain technologies are most beneficial. The authors in [16] carried out a comprehensive overview of Blockchain technologies in the framework of 140 Blockchain projects and startups. The study [23] reviews a number of projects on Blockchain applications. It shows that the support of policymakers in terms of the development of technical standards and guidance and the organization of sandboxes is crucial for the promotion of Blockchain technology in the electric power industry. The most recent study [24] looks at the results of Blockchain's introduction into the energy sector within the context of the German energy transition.

Recent papers examine the possibilities to combine various digital tools to improve the performance of the energy market [25–27] or attempt to evaluate the effects of the implementation of digital techniques [17,18,28,29]. The paper [25] explores the advantages of Cloud Computing in interconnection with the development of Smart Grid solutions for the energy sector. The paper suggests that a transformation to Cloud Computing allows

the achievement of the ultimate goals of Smart Grid implementation, such as energy cost reduction and two-way communication. The paper [26] presents a qualitative overview of the Big Data and IoT implications for the business. The authors in [27] describe the opportunities to use Blockchain technology in IoT-based smart grid monitoring.

The survey [29] suggests a qualitative approach to the evaluation of different digital energy services based on decision factors such as technical feasibility, business potential, behavioral change, and innovativeness. The authors argue that behavioral and business factors have the most effects on the decision-making process. They also show that the interrelation of the digital tools makes it logical to consider the combination of the digital instruments in the form of a digital platform. The paper [18] concerns Smart Grids in the context of the Internet of Things (IoT) concept and evaluates the risks of data privacy and security. The study [17] evaluates the viability of further rolling out a Blockchain technology in Japan from a technological, economic, social, environmental, and institutional perspective. The authors argue that the future of Blockchain-based energy systems heavily depends on a comprehensive approach that encompasses the mentioned factors. The authors of [28] show that among such digital tools as Artificial Intelligence, Big Data, Internet of Things, Robotics, Blockchain technology, and Cloud computing, AI has the most valuable effect on a company's performance, as well as on the overall wage level within the company and in the local job market.

Figure 1 shows the main directions of the implementation of digital technologies in the Russian electric power industry.

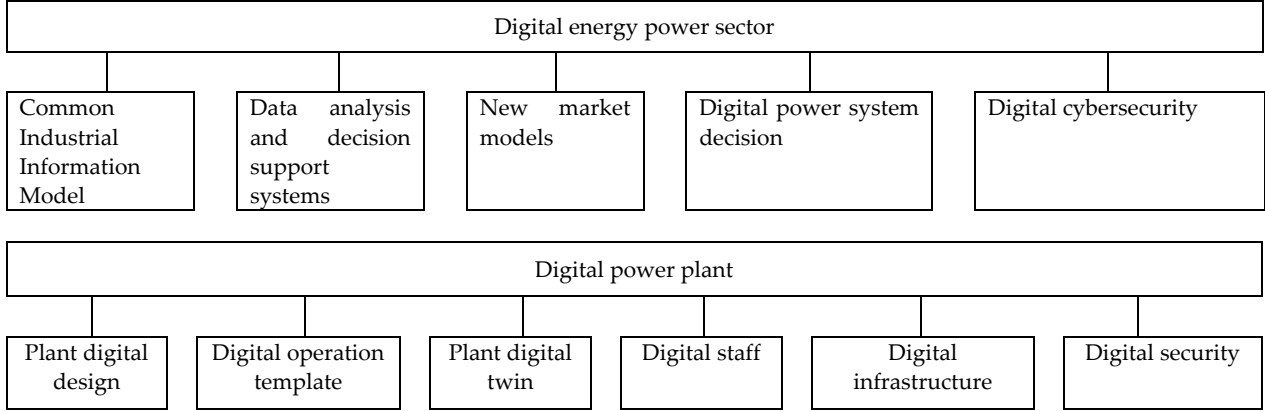

**Figure 1.** Main directions of the implementation of digital technologies in the Russian electric power industry. Source: compiled by authors based on [30].

A detailed description of each direction presented above is given in [31].

## 2. Materials and Methods

The methodology applied in the present research is based on the authors' previous findings, available on request [32]. In their earlier work, the scholars explored the causal relationships between sales revenue and EBIT received by a nuclear power plant (NPP), thermal power plant (TPP), and hydroelectric power plant (HPP), and production factors: material costs, including fuel costs; wage costs, including mandatory pension contributions, healthcare and social deductions, depreciation deductions, and miscellaneous costs.

The results show the following:

- An increase in material costs leads to an increase in sales revenue for NPP and HPP but a decrease in sales revenue for TPP;
- An increase in depreciation deductions causes the sales revenue and EBIT to rise;
- An increase in miscellaneous costs leads to a decrease in EBIT for NPP and HPP but a growth in sales revenue and EBIT for TPP.

The results obtained should be taken into account prior to choosing digital technologies and considering the direction of their implementation.

The main notation used in the research is presented in Table 1.

**Table 1.** Notation.

| Notation. | Definition |
|---|---|
| $C_{OPEX}^1$ and $C_{OPEX}^2$, [rub] | Energy production costs before and after implementation of digital technology |
| $P_1$ и $P_2$, $\left(\frac{rub}{kW\cdot h}\right)$ | Costs of energy sold before and after implementation of digital technology (The Russian electricity market can be conventionally divided into regulated (95%) and open (5%). The electricity price in the regulated market is set by the state (in 95% of cases, $P_1 = P_2$). Thus, economic efficiency of the implementation of digitals tool due to the increase in energy sales revenue is created by the volume of energy produced and then sold.) |
| $Q_1$ and $Q_2$, $[kW\cdot h]$ | Energy production before and after implementation of digital technology |
| $R^1 = P_1 Q_1$, [rub] $R^2 = P_2 Q_2$, [rub] | Energy sales revenue before and after implementation of digital technology |
| $PR^1 = R^1 - C_{OPEX}^1$, [rub] $PR^2 = R^2 - C_{OPEX}^2$, [rub] | Energy sales profit before and after implementation of digital technology |

Source: compiled by authors.

It is possible to identify several types of efficiency associated with the implementation of digital tools:

1. Economic efficiency of the implementation of digital tools due to the decrease in energy production costs $(E^C)$, [rub] is present if $C_{OPEX}^1 > C_{OPEX}^2$, when $R^1 = R^2$, then $PR^1 < PR^2$. In this case, $PR^2 - PR^1 = E^C$. In order to determine the economic efficiency of the implementation of digital tools due to the decrease in energy production costs, we need to solve the following optimization problem: $\left\{E^C \rightarrow \max \mid C_{OPEX} \rightarrow \min\right\}$;

2. Economic efficiency of the implementation of digital tools due to the increase in energy sales revenue $(E^R)$, [rub] is present if $R^2 > R^1$, when $C_{OPEX}^1 = C_{OPEX}^2$, then $PR^1 < PR^2$. In this case, $PR^2 - PR^1 = E^R$. In order to determine the economic efficiency of the implementation of digital tools due to the increase in energy sales revenue, we need to solve the following optimization problem: $\left\{E^R \rightarrow \max \mid R \rightarrow \max\right\}$;

3. Aggregated economic efficiency due to the changes in energy production costs as well as energy sales revenue $(E^I)$, [rub] is present if $PR^2 - PR^1 = E^I > 0$, when $\frac{\partial PR^2}{PR^2} > \frac{\partial PR^1}{PR^1}$. That is, the energy sales profit growth rate after the implementation of digital technology is higher than the energy sales profit growth rate before the implementation of digital technology. In this case, $\left\{E^I \rightarrow \max \mid C_{OPEX} \rightarrow \min, R \rightarrow \max\right\}$.

4. Social efficiency $(E^S)$ of the implementation of digital tool due to the reduction in mortality, GHG emissions, the increase in energy availability, etc.

The social efficiency of digital technology implementation is a complex concept. Numerous studies on social efficiency do not provide an answer to the question "what is social efficiency?". We refer to the definition of the social state from [33], where all material goods are distributed in such a way as to improve the quality and raise the standard of living of its citizens. Thus, in our case, social efficiency occurs if the quality and standard of living increases, which can be evaluated either quantitatively or qualitatively. In this study, the authors performed a qualitative assessment of social efficiency in a structured manner, as presented in Figure 2.

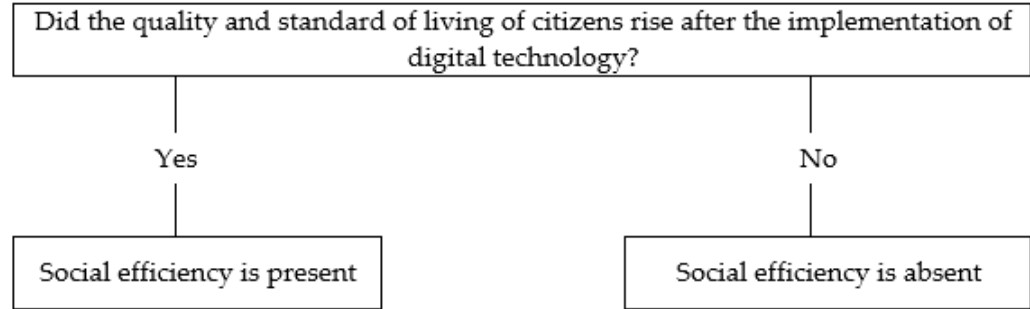

**Figure 2.** Diagram for assessment of social efficiency. Source: compiled by authors.

Further, the authors identify the following cases:

- Economic efficiency of the implementation of digital technology is absent ($E^C < 0$, $E^R < 0$, $E^I < 0$), and social efficiency is absent ($E^S < 0$). In this case, the implementation of digital technology is not feasible from both economic and social perspectives.
- Economic efficiency of the implementation of digital technology is absent ($E^C < 0$, $E^R < 0$, $E^I < 0$), and social efficiency is present ($E^S > 0$)). In this case, it is necessary to assess the social effects and the possibility of its commercialization—for example, by receiving government grants, subsidies, tax benefits, etc.
- Economic efficiency of the implementation of digital technology is zero ($E^C = 0$, $E^R = 0$, $E^I = 0$), and social efficiency is present ($E^S > 0$). In this case, the implementation of digital technology is feasible since it leads to the improvement of the quality and standard of living of citizens. The authors argue that if the social effect of the implementation of digital tools is due to a reduction in mortality and harm to human health, then the digital technology should undoubtedly be implemented for ethical reasons. This ethical principle has been implemented in Russian electric power companies.

The economic efficiency of the implementation of digital tools due to the increase in energy sales revenue, as well as the aggregated economic efficiency and social efficiency, are subjects for a separate study as they lie beyond the scope of the present research. The scientific goal of this paper is to evaluate the economic efficiency of the implementation of digital tools due to the decrease in energy production costs. The evaluation is performed taking into account:

The energy cost calculation method before and after the implementation of digital technology ($C_{OPEX}$);

The digital technology cost calculation method ($C^{DT}$) based on the technology readiness level ($k$).

### 2.1. Energy Cost Calculation Method before Implementation of Digital Technology

Operating expenses (OPEX) of electricity generation consist of depreciation deductions, material costs, wage costs of operation and maintenance personnel, mandatory pension contributions, healthcare and social deductions, equipment operation and maintenance costs, as well as miscellaneous costs, overhead, and non-production costs.

Operating expenses allow us to assess the economic efficiency of a certain type of electricity generation—a nuclear power plant (NPP), thermal power plant (TPP), hydroelectric power plant (HPP), or plants based on renewable energy sources (RES).

The annual operating expenses of any type of power generation are calculated using the following notation:

$$C_{OPEX} = C_{dep} + C_M + C_w + C_{soc} + C_E + C_{mc} + C_{oc} + C_{nm} \tag{1}$$

where $C_{dep}$ is the depreciation deductions, [*rubles*]; $C_M$ is the material costs, [*rubles*]; $C_W$ is the wage costs of operation and maintenance personnel, [*rubles*]; $C_{soc}$ is the mandatory

pension contributions, healthcare, and social deductions, [*rubles*]; $C_E$ is the equipment operation and maintenance costs, [*rubles*]; $C_{mc}$ is the miscellaneous costs, [*rubles*]; $C_{oc}$ is the overhead costs, [*rubles*]; and $C_{nc}$ is the non-production costs, [*rubles*].

The specific formation of each component of OPEX is determined by the type of power generation and is described in Appendix B.

Thus, the annual operating expenses before the implementation of digital technology are expressed as follows

$$C^1_{OPEX} = C^1_{dep} + C^1_M + C^1_w + C^1_{soc} + C^1_E + C^1_{mc} + C^1_{oc} + C^1_{nm} \tag{2}$$

### 2.2. Digital Technology Cost Calculation Method

As soon as an electric power company makes a decision about the implementation of a digital technology, it conducts R&D on its own (in-house R&D) or purchases the results of R&D (external R&D). At the end of the R&D phase, the digital technology becomes either an intangible asset or a tangible asset developed by the company (in-house development) or purchased from suppliers (external purchase). Hence, the costs per one digital technology ($C^{DT}$) are determined as follows in Table 2.

**Table 2.** Digital technology costs.

| $C^{DT}, \frac{rub.}{unit} \times unit = rub.$ | R&D | Intangible Assets | Tangible Assets |
|---|---|---|---|
| In-house | $C^{R\&D}_{OPEX}$ $= C_{dep} + C_M + C_w$ $+ C_{soc} + C_E + C_{mc}$ $+ C_{oc}$ | $C^{IA}_{OPEX}$ $= C_{dep} + C_M + C_w$ $+ C_{soc} + C_{mc} + C_{oc}$ | $C^{TA}_{OPEX} = C_{dep} + C_M +$ $C_w + C_{soc} + C_E +$ $C_{mc} + C_{oc}$ |
| External | $C^{R\&D}_{TCO}$ | $C^{IA}_{TCO}$ | $C^{TA}_{TCO}$ |

Source: compiled by authors.

The last line of Table 2 requires some comments. When an electric power company purchases the results of R&D, the cost of R&D ($C^{R\&D}_{TCO}$) is an exogenous variable determined by the supplier.

When an electric power company purchases the digital technology as an intangible asset (for example, software), the cost of the digital technology ($C^{IA}_{TCO}$) consists of costs to obtain the exclusive/non-exclusive rights to the intangible asset, payments of duties and taxes, and others. When an electric power company purchases the digital technology as a tangible asset (for example, a smart video camera), the cost of the digital technology is $C^{TA}_{TCO}$.

$C^{R\&D}_{TCO}$, $C^{IA}_{TCO}$, $C^{TA}_{TCO}$ also include the share of profit of the digital technology's developer. If the digital technology is a defense or a military product, then the share of profit is not incorporated into the cost.

Moreover, in accordance with accounting and financial analysis, the components of $C^{DT}$ are appropriately attributed among the components of $C_{OPEX}$. The economic efficiency is higher if $C^{DT}$ are attributed considering how changes in components affect the profit, as shown above [32]. Table 3 shows the attribution of $C^{DT}$ components among $C_{OPEX}$ components.

**Table 3.** The attribution of $C^{DT}$ components between $C_{OPEX}$ components.

| Cost Attribution | R&D | Intangible Assets | Tangible Assets |
|---|---|---|---|
| In-house development | The costs are covered from R&D funds (% from revenue) and included in miscellaneous costs | The costs are attributed and recognized in the relevant costs | The costs are attributed and recognized in the relevant costs |
| Purchase | The costs are attributed and recognized in miscellaneous costs as service of third party | The costs are attributed and recognized in miscellaneous costs as service of third party and other costs as miscellaneous costs | The costs are attributed and recognized in miscellaneous costs as service of third party and other costs as miscellaneous costs |

Source: compiled by authors.

As mentioned earlier, the cost of digital technology $(C^{DT})$ depends, among other factors, on the technology readiness level: the lower the technology readiness level, the higher the cost.

The cost of digital technology $(C^{DT_i})$ considering the technology readiness level is defined as follows:

$$C^{DT_i} = C^{DT}/k \tag{3}$$

where $i$ is the phase or technology readiness level (see Appendix A), and $k$ is the coefficient of technology readiness such that $k \in (0, 1]$ and if $k \to 0$, $C^{DT_i} \to \infty$.

The value of $k$ corresponds to the share of government financing for each phase of technology readiness. According to [34], the recommended shares of government funds are as shown in Table 4.

**Table 4.** Recommended share of government funds for each phase of technology readiness.

| Directions | Phase/Level | Share of Government Funds |
|---|---|---|
| Fundamental research | 0 | 100% |
| | 1 | 90–100% |
| Applied research | 2 | 80–90% |
| | 3 | 70–80% |
| Experimental development | 4 | 60–70% |
| | 5 | 40–60% |
| | 6 | 40–60% |
| | 7 | 30–40% |
| Pilot production | 8 | 10–30% |
| Industrial production | 9 | 0–10% |

Source: compiled by authors based on the report of the Ministry of Science and Higher Education of the Russian Federation, 2022.

In this case, $k$ is calculated using the following equation:

$$k = \frac{(100 - x_i)}{100} \tag{4}$$

where $x_i$ is the funding raised at phase $i$ (%). In order to avoid division by zero, it is necessary to assume that $x_i < 100\%$.

As an example, let us determine the cost of digital technology with technology readiness level 0, 4, and 9.

$$
\begin{aligned}
C_0^{DT} &= C^{DT} / \frac{(100-x_i)}{100} = C^{DT} / \left( \frac{(100-99.99)}{100} \right) = 10000 C^{DT}, \\
C_4^{DT} &= C^{DT} / \frac{(100-x_i)}{100} = C^{DT} / \frac{(100-60)}{100} = 2.5 C^{DT}, \\
C_9^{DT} &= C^{DT} / \frac{(100-x_i)}{100} = C^{DT} / \frac{(100-0)}{100} = C^{DT}.
\end{aligned}
\tag{5}
$$

### 2.3. Energy Cost Calculation Method after Implementation of Digital Technology

Annual operating expenses after the implementation of digital technology are calculated as follows:

$$
C_{OPEX}^2 = C_{dep}^2 + C_M^2 + C_w^2 + C_{soc}^2 + C_E^2 + C_{mc}^2 + C_{oc}^2 + C_{nm}^2
\tag{6}
$$

where $C_{dep}^2 = C_{dep}^1 + \alpha_1 C^{DT}$, $\alpha_1$ is a share of the digital technology cost attributed to depreciation deductions, [*rubles*]; $C_M^2 = C_M^1 + \alpha_2 C^{DT}$, $\alpha_2$ is a share of the digital technology cost attributed to material costs, [*rubles*]; $C_w^2 = C_w^1 + \alpha_3 C^{DT}$, $\alpha_3$ is a share of the digital technology cost attributed to wage costs, [*rubles*]; $C_{soc}^2 = C_{soc}^1 + \alpha_4 C^{DT}$, $\alpha_4$ is a share of the digital technology cost attributed to mandatory pension contributions, healthcare, and social deductions, [*rubles*]; $C_E^2 = C_E^1 + \alpha_5 C^{DT}$, $\alpha_5$ is a share of the digital technology cost attributed to equipment operation and maintenance costs, [*rubles*]; $C_{mc}^2 = C_{mc}^1 + \alpha_6 C^{DT}$, $\alpha_6$ is a share of the digital technology cost attributed to miscellaneous costs, [*rubles*]; $C_{oc}^2 = C_{oc}^1 + \alpha_7 C^{DT}$, $\alpha_7$ is a share of the digital technology cost attributed to overhead costs, [*rubles*]; and $C_{nm}^2 = C_{nm}^1 + \alpha_8 C^{DT}$, $\alpha_8$ is a share of the digital technology cost attributed to non-production costs, [*rubles*].

Thus, $\alpha_1 + \alpha_2 + \alpha_3 + \alpha_4 + \alpha_5 + \alpha_6 + \alpha_7 + \alpha_8 = 1$.

Hence, considering the technology readiness level, the annual operating expenses after implementation equals:

$$
C_{OPEX}^2 = \left( C_{dep}^2 + C_M^2 + C_w^2 + C_{soc}^2 + C_E^2 + C_{mc}^2 + C_{oc}^2 + C_{nm}^2 \right) \cdot \frac{1}{k}
\tag{7}
$$

It would seem logical to transfer the notation above as follows:

$$
C_{OPEX}^2 = \left( C_{OPEX}^1 + C^{DT} (\alpha_1 + \alpha_2 + \alpha_3 + \alpha_4 + \alpha_5 + \alpha_6 + \alpha_7 + \alpha_8) \right) \times \frac{1}{k} = \left( C_{OPEX}^1 + C^{DT} \right) \frac{1}{k}
\tag{8}
$$

However, in this case, the coefficients $\alpha_1 \ldots \alpha_8$ do not affect the economic efficiency of the implementation of the digital technology, although, previously, we have shown the opposite. Therefore, it is necessary to use the coefficients $\alpha_1 \ldots \alpha_8$ for the evaluation of the economic efficiency of the implementation of the digital technology.

### 2.4. Economic Efficiency of the Implementation of Digital Technology Due to the Decrease in Energy Production Cost

The economic efficiency of the implementation of digital technology due to the decrease in energy production costs equals

$E^C = PR^2 - PR^1 > 0$, because $PR^1 = R^1 - C_{OPEX}^1$, $PR^2 = R^2 - C_{OPEX}^2$, and $R^1 = R^2$, then $E^C = R^2 - C_{OPEX}^2 - R^1 + C_{OPEX}^1 > 0$ or $E^C = C_{OPEX}^1 - C_{OPEX}^2 > 0$.

Since the coefficients $\alpha_1 \ldots \alpha_8$ are rather significant, in our opinion, the economic efficiency should be evaluated by the change in each component of the electricity cost, i.e., $\frac{dC_{OPEX}^2}{dC_{dep}^2}$, $\frac{dC_{OPEX}^2}{dC_M^2}$, $\frac{dC_{OPEX}^2}{dC_w^2}$, $\frac{dC_{OPEX}^2}{dC_{soc}^2}$, $\frac{dC_{OPEX}^2}{dC_E^2}$, $\frac{dC_{OPEX}^2}{dC_{mc}^2}$, $\frac{dC_{OPEX}^2}{dC_{oc}^2}$, $\frac{dC_{OPEX}^2}{dC_{nm}^2}$ and $\frac{dC_{OPEX}^1}{dC_{dep}^1}$, $\frac{dC_{OPEX}^1}{dC_M^1}$, $\frac{dC_{OPEX}^1}{dC_w^1}$, $\frac{dC_{OPEX}^1}{dC_{soc}^1}$, $\frac{dC_{OPEX}^1}{dC_E^1}$, $\frac{dC_{OPEX}^1}{dC_{mc}^1}$, $\frac{dC_{OPEX}^1}{dC_{oc}^1}$, $\frac{dC_{OPEX}^1}{dC_{nm}^1}$.

The Results and Discussion section illustrates the practical application of the methods described above through the calculation of the economic efficiency of the implementation of a particular digital technology.

## 3. Results and Discussions

In order to proceed further, it is necessary to make some assumptions. We consider the nuclear energy and an NPP as a generating facility where we have implemented a predictive model of equipment. A predictive model of equipment refers to the end-to-end technology "New Manufacturing Technologies" (Appendix A), namely, digital design–digital shadow. It uses instruments of predictive analytics. Predictive analytics is a set of equipment's physical and statistical models that describe the behavior of the equipment in different modes and operating conditions.

A predictive model allows us to achieve the following targets:

To reduce the duration of outages by determining the short-list of equipment to be repaired and the scope of repairs;

To reduce the duration of unplanned shutdowns by improving the reliability of equipment and resource requirement planning (materials, spare parts, personnel);

To increase the NPP's availability factors (as a consequence of the first two points);

To increase the NPP's safety indicators by decreasing personnel exposure doses and improving the control over equipment and systems life by reducing repair cycles;

To improve the NPP's economic performance by reducing maintenance and repair costs and duration of outages and by increasing power generation.

In the context of the declared transition from 18-month to 24-month fuel cycle (750 effective days), the appropriate choice of equipment for planned predictive maintenance given a full compliance with safety requirements becomes crucial.

The Novovoronezh NPP was selected as a pilot plant for the implementation of predictive analytics [35].

Currently there are several thermal and vibration models in operational testing: for a turbine generator, for a turbine unit, and a for circulation pump. Present research evaluates the cost-effectiveness of introduction of a predictive model for a turbine generator which estimates the change in its parameters over time and compares with "ideal" values. If the actual parameters deviate by more than 5% from the "ideal" parameters set by the model, this deviation information is sent to the operator. The deviation value of 5% is set by the developer of the model and differs from the limits of tolerances set by the developer of the turbine generator. The values 5% and 10% are conventional values equal to the standard deviation according to the random values distribution law of a controlled parameter. The intercorrelation between parameters (e.g., temperature and pressure) is also taken into consideration. Hence, the deviation has three coordinates (X, Y, T), where X and Y are matrices of parameters values in time T. Each coordinate has a number (e.g., 15,483), which is the deviation number. The operator analyzes each deviation once a week and sends a report to the developer of the model. Now, on average, there are 10–15 deviations per week. The results of the deviation analysis can be used to make the decisions between the evaluation of technical condition of the equipment and model adjustment.

The predictive model has the following tasks:

To determine the probability of the turbine generator's no-failure operation during a specified time interval—in our case, the operating time of planned preventive maintenance. For example, what is the probability that turbine generator will be in operation for 18,000 h?

To determine a time interval during which the turbogenerator will not fail with a given probability. For example, for how long a turbine generator will work without failures with a probability of 99.99(9)%?

The implementation of the predictive model of the turbine generator allows us to achieve control over equipment life as well as to plan preventive maintenance.

At the present time, the NPPs' turbine generators are subject to planned preventive maintenance. However, there is a prospect for transition to condition-based maintenance. Table 5 shows frequency and duration of outages for VVER-1200.

**Table 5.** Frequency and duration of outages for VVER-1200.

| Type of Outage | Major | Medium |
|---|---|---|
| Frequency, month number | 6, 42, 78, 114 | 24, 60, 96, 132 |
| Actual duration, days | 45 | 32 |
| Prospective duration, days | 40 | 30 |

Source: [36].

The statistics of unplanned shutdowns due to equipment and turbine generator failures at the nuclear power plant are given in Table 6.

**Table 6.** Number and duration of unplanned shutdowns due to equipment failures of power units and turbine generators at NPPs in 2015–2021.

| Period | Amount of Power Units and Turbine Generators Shutdowns, Units | | Duration of Power Units and Turbine Generators Shutdowns, Days | |
|---|---|---|---|---|
| | Power Units | Turbine Generators | Power Units | Turbine Generators |
| 2015 | 34 | 20 | 105.1 | 84.4 |
| 2016 | 29 | 21 | 189 | 63.2 |
| 2017 | 28 | 28 | 149.6 | 129.9 |
| 2018 | 27 | 19 | 186 | 101.8 |
| 2019 | 28 | 27 | 126.9 | 55.9 |
| 2020 | 19 | 17 | 52.9 | 32.4 |
| 2021 | 27 | 21 | 183.6 | 59 |
| Total | 192 | 153 | 993.1 | 526.6 |
| Medium | 27 | 22 | 141.9 | 75.2 |

Source: compiled by authors.

As shown in Table 6, the Russian NPPs will, on average, reduce unplanned showdowns by 75.2 days per year, which will allow us to:

- Generate extra electric power;
- Avoid penalties imposed by the system operators at the electricity market;
- Reduce the cost of electricity due to savings in repairs;
- Reduce doses of irradiation for maintenance personnel (Table 7);
- Reduce greenhouse gas emissions due to nuclear generation without using fossil fuels, etc. (the results of research in this area will be presented in future articles of the authors).

**Table 7.** Novovoronezh NPP's personnel exposure doses during planned preventive maintenance.

| Period | Power Units, N | Type of Repair | Exposure Doses, mSv | Duration of Repair, Days |
|---|---|---|---|---|
| 2015 | 3/4/5 | MiR/MaR, M/MaR | 3.64 | 166.5 |
| 2016 | 3/4/5 | MiR/MiR/MiR | 1.86 | 98.5 |
| 2017 | 4/5/6/6 | MaR, SLE/MiR/RM/MiR, M | 0.34 | 129 |
| 2018 | 4/5/6 | MaR, SLE/MiR/MiR, M | 3.94 | 450.5 |
| 2019 | 4/5/6 | MiR/MaR/MaR | 0.6 | 179 |
| 2020 | 4/5/6/7 | MiR/MiR/MiR/MiR | 1.84 | 185 |

Note: Major (MaR), medium (MiR), current (CR), routine maintenance (RM), service life extension (SLE), modernization (M). Source: compiled by authors based on [37].

Thus, implementation of a predictive model obviously has a positive social effect and, presumably, economic effect.

The Novovoronezh NPP's personnel exposure doses during planned preventive maintenance are given in Table 7. Both the type and labor intensity of repairs influence the exposure doses.

The NPPs' energy availability and load factors are given in the PRIS IAEA database [35]. The Novovoronezh NPP's energy availability factor and load factor are given in Table 8.

**Table 8.** Novovoronezh NPP's energy availability factor and load factor (VVER-1200), %.

| Period | Power Unit No.6 | | Power Unit No.7 | |
|---|---|---|---|---|
| | Energy Availability Factor | Load Factor | Energy Availability Factor | Load Factor |
| 2017 | 62.4 | 60.9 | - | - |
| 2018 | 81.3 | 79.4 | - | - |
| 2019 | 76.7 | 74.6 | 85.5 | 83.5 |
| 2020 | 80.3 | 78.7 | 82.1 | 78.6 |
| 2021 | 78.8 | 78.1 | 79.6 | 76.2 |
| Medium | 76.3 | 74.8 | 79.6 | 77.9 |

Source: [35].

Considering the significant volume of initial data on repairs for turbine generators of VVER-1200, we explore a particular case—the repair of turbine generator TZV-1200-2AU3 (ТЗВ-1200-2АУ3 in Russian), with input data given in Table 9.

**Table 9.** Initial data on repairs for turbine generator TZV-1200-2AU.

| | MiR | MaR |
|---|---|---|
| Labor intensity, man-hour | 5934 | 7998 |
| Labor cost, mln rub. | 4.2 | 5.7 |
| Material cost, mln rub. | 0.1 | 0.1 |
| Fitting cost, mln rub. | 0 (We assume that fittings are available at the NPP) | 0 |
| Duration, days | 26 | 35 |

Source: compiled by authors.

The turbine generator has first class of safety, and therefore, in accordance with regulatory requirements, a medium repair is carried out once a year, with a major one once every 4.5 years.

The predictive model for the Novovoronezh NPP's turbine generator is an in-house product. It was developed in cooperation with JSC "All-Russian Research Institute for Nuclear Power Plants Operation" and JSC "Konsist-OS" using new production technologies, including a digital twin and a digital shadow of the turbine generator. Both technologies have readiness level 9 in Russia; hence, $C_9^{DT} = C^{DT} / \frac{(100-x_i)}{100} = C^{DT} / \frac{(100-0)}{100} = C^{DT}$.

Then, $C^{DT} = C_{OPEX}^{IA} = C_{dep} + C_M + C_w + C_{soc} + C_{mc} + C_{oc}$, where $C^{DT} = 43$ *mln rub.*, $\alpha_1 = 0.3$, $\alpha_3 = 0.33$, $\alpha_4 = 0.11$, $\alpha_6 = 0.06$, $\alpha_7 = 0.2$, $\alpha_2, \alpha_5, \alpha_8 = 0$.

Thus, the economic efficiency of the implementation of the predicative model is present if $E^C = C_{OPEX}^1 - C_{OPEX}^2 > 0$, given $C_{OPEX}^1 = C_{dep}^1 + C_M^1 + C_w^1 + C_{soc}^1 + C_E^1 + C_{mc}^1 + C_{oc}^1 + C_{nm}^1$, $C_{OPEX}^2 = \left( C_{dep}^1 + 0.3C^{DT} \right) + C_M^1 + \left( C_w^1 + 0.33C^{DT} \right) + \left( C_{IPP}^1 + 0.11C^{DT} \right) + C_E^1 + \left( C_{mc}^1 + 0.06C^{DT} \right) + \left( C_{oc}^1 + 0.2C^{DT} \right) + C_{nm}^1$.

Let us discuss the evaluation of the economic efficiency of the implementation of the predictive model of the turbine generator for the following cases:

- Prevention of unplanned shutdowns due to turbine generator failure;
- Transition from planned preventive maintenance to condition-based maintenance.

In the first case, there is a power unit unplanned shutdown every two years on average, and the unplanned repair duration equals 3.4 days. Using data from Table 9 above, the cost of repair is $\frac{5.7 \ mln. \ rub.}{35 \ days} \times 3.4 \ days \div 2 \ years = 0.275 \ mln.rub.$ Thus, even roughly, calculations show that the energy cost will be reduced by 0.275 $mln.rub.$ due to the decrease in $C_w^1$ and $C_{soc}^1$. Considering the predictive model cost, the economic efficiency is present if $E^C = \left(C_{OPEX}^1 - 0.275\right) - \left(C_{OPEX}^2 + 43\right) > 0$ (without considering the effect of coefficient $\alpha$).

We also obtain economic efficiency due to the additional power generation, which equals $1200 \ MW \cdot \frac{3.4 \ days}{2} \cdot 24h = 48960 \ MWh$ under nominal capacity operation mode.

In summary, the estimated economic efficiency of the implementation of the predictive model of the turbine generator demonstrates the advantages of energy power digitalization worldwide, including the achievement of the declared goals of sustainable development and the transition to low-carbon energy discussed in the Introduction.

## 4. Conclusions

The scientific results obtained are valuable for scholars carrying out research in the area of the efficiency of economic digitalization, since they hold for all countries and types of energy generation.

The most precise type of efficiency obtained from the implementation of digital technology is the aggregate economic efficiency. However, it requires a large amount of statistical data, for at least a five-year period, on the following:

- The components of energy costs by type of energy generation;
- The components of digital technology costs;
- The volume of electricity production;
- Electricity tariffs.

The results of assessment of the aggregate economic efficiency will be presented in the authors' future work.

The directions of further research are the following:

- The development of an analytical notation to show that the digital technology cost depends on the speed and volume of its distribution, as well as the technology readiness level, the level of technological development of material and technical resources in the industry and in the country, and labor and capital availability. The cross-influence of industries is evident. Moreover, the authors have already obtained the analytical dependence of the cost of digital technology on the technology readiness level ($k$), volume of distribution ($V$) and speed of distribution ($S$) of digital technology, and the effect of the technological scale ($A$).
- Conducting comprehensive calculations for the evaluation of the economic efficiency of the implementation of the turbine in the electric power industry in Russia, taking into account the components of energy costs and the $\alpha$-cofficients.
- Estimation of the elasticity coefficients for the components of digital technology and energy cost. The obtained results will allow the planning of scientific and technological progress in the energy power industry, which would provide the state with a reasonable allocation of resources.

The authors stand by the necessity to plan the technological progress rather than predict it or rely on a casual event.

**Author Contributions:** Conceptualization, V.G. and M.V.; methodology, M.V.; software, V.G.; writing—original draft preparation, V.G.; writing—review and editing, V.G. and M.V.; visualization, V.G. and M.V. All authors have read and agreed to the published version of the manuscript.

**Funding:** This research received no external funding.

**Data Availability Statement:** Data supporting reported results can be found using the following links: https://digital.gov.ru/ru/documents/6650/ (accessed on 1 May 2022), https://eng.rosstat.gov.ru/sdg/reporting-status (accessed on 1 May 2022), https://world-nuclear.org/ (accessed on 1 May 2022), https://fcpir.ru/upload/medialibrary/955/gt_57_14vn_metodika-ugt-_002_.pdf (accessed on 1 May 2022).

**Conflicts of Interest:** The authors declare no conflict of interest.

## Appendix A. List of End-to-End Technologies and Sub-Technologies by Technological Readiness Level (TRL)

| Name of Technology | Sub-Technology | TRL World | TRL Russia |
|---|---|---|---|
| Wireless communication technologies | WAN (Wide Area Network) 5G/LTE | 8/9 | 3/4 |
| | LPWAN (Low Power Wide Area Network) | 9 | 9 |
| | WLAN (Wireless Local Area Network) Li-Fi/Wi-Fi | 8/9 | 6/7 |
| | PAN (Personal Area Network) | 9 | 9 |
| | Satellite Communication Technologies Satellite Broadband/Satellite Internet of Things/Satellite Personal Communications | 9/2/6 | 5/2/5 |
| Distributed registry systems | Data integrity and consistency technologies (consensus) | 8 | 8 |
| | Technologies for creating and executing decentralized applications and smart contracts | 7 | 6 |
| | Data organization and synchronization technologies | 8 | 7 |
| Robotics and sensorics components | Sensors and digital robotic kits' components of for human–machine interaction | 7 | 7 |
| | Sensor-motor coordination and spatial positioning technologies | 6 | 9 |
| | Sensors and sensory information processing | 6 | 9 |
| New production technologies | Digital design, mathematical modeling, and product life cycle management (Smart Design) | 7–9 | 6–9 |
| | Smart manufacturing technologies | 6–7 | 4–5 |
| | Manipulators and manipulation technologies | 6 | 9 |
| Neurotechnology and artificial intelligence | Computer vision | 6 | 6 |
| | Natural language processing | 6 | 6 |
| | Speech recognition and synthesis | 5 | 5 |
| | Recommendation systems and intelligent decision support systems | 7 | 5 |
| | Advanced AI methods and technologies | 2 | 2 |
| | Neuroprocessing | 5 | 5 |
| | Neurointerfaces, neurostimulation and neurosensing | 3 | 3 |
| Quantum technologies | Quantum computing | 4–5 | 3–4 |
| | Quantum communications | 9 | 8 |
| | Quantum sensors and metrology | 3–9 | 1–5 |

| Name of Technology | Sub-Technology | TRL World | TRL Russia |
|---|---|---|---|
| Virtual and augmented reality technologies | VR/AR content development tools and developer side user experience (UX) enhancement technologies | 9 | 8 |
| | Platform solutions for users: editors of content creation and distribution | 7 | 6 |
| | Motion capture technologies in VR/AR and photogrammetry | 9 | 7 |
| | Feedback interfaces and sensors for VR/AR | 7 | 6 |
| | Graphical output technologies | 9 | 7 |
| | Data optimization technologies for VR/AR | 9 | 5 |

Source: compiled by authors based on reports of the Ministry of Digital Development, Communications and Mass Media of the Russian Federation, 2022.

## Appendix B. Components of Electricity Costs

Depreciation deductions ($C_{dep}$) are calculated as follows:

$$C_{dep} = K \sum_{i=1}^{n} \frac{E_i \cdot s_{dep\_i}}{100}, \ [rubles] \tag{A1}$$

where $K$ is the balance costs of fixed assets, $[rubles]$; $s_{dep\_i}$ is the share of equipment type $i$ in the balance costs of fixed assets; and $E_i$ is the rate of depreciation, % per year.

Material costs ($C_M$) are operation costs, which include the expenses of fuel for energy production, raw materials, spare parts, instruments and devices determined by the design features of the plant, production processes, as well as expenditure rates for the materials and spare parts approved by the state, industry, or plant.

Fuel costs for electricity and heat generation depend on the type of generation and the features and properties of the fuel used (fossil, nuclear, solar, tidal, wind, etc.), such as calorific value, ballast and hazardous impurity content, radioactive waste generation, and others.

In the present paper, the authors employ the cost calculation method for the fuel component of energy produced by TPP and NPP, which is described in full detail in [38–40]. It is apparent that the calculation of the fuel costs of energy produced by HPP and RES has not yet been performed. The material expenses of HPP and RES are limited to repair and operational needs.

The wage costs of operation and maintenance personnel ($C_W$) are calculated using the following notation:

$$C_W = C_w^O + C_w^{M\&R}, \ [rubles] \tag{A2}$$

where $C_w^O$ is the wage costs of operation personnel, $[rubles]$; and $C_w^{M\&R}$ is the wage costs of maintenance personnel, $[rubles]$.

Operation and maintenance personnel labor remuneration comprises basic and supplement payments, as well as compensation payments. The basic part depends on the employee's pay rate, the average monthly output in hours, bonuses, incentive allowances, and other payments related to the KPIs and overtime. Compensation payments are related to work in difficult, harmful, and (or) dangerous working conditions and are calculated as a percentage of the employee's pay rate. Supplement payments are determined as a percentage of basic and compensation payments. They include payments indirectly related to the performance of the main work duties, such as mentoring or the implementation of socially important activities (maternity, donations, etc.). Maintenance personnel labor remuneration is calculated using multipliers related to the specifics of the repair work, which is explained in [41].

Mandatory pension contributions, healthcare, and social deductions ($C_{soc}$) are calculated in accordance with current legal requirements.

Equipment operation and maintenance costs ($C_E$) are calculated based on the cost of machine hours for the equipment and tools used to execute repairs. Equipment operation and maintenance costs are considered a separate term in the case of work performed by contractors using their own equipment. If a contractor performs works using plant equipment, the costs are included in the overhead.

Miscellaneous costs ($C_{mc}$) depend on the electricity generation type and consist of industry reserves, expenditure on third-party services, business travel, transportation, and others. Industry reserves are formed mainly by NPP to ensure nuclear, radiation, technical, and fire safety during NPP operation; the security, accounting, and control of nuclear materials; the development of nuclear plants; decommissioning, and the disposal of radioactive waste.

Overhead costs ($C_{oc}$) are expenses at the unit level ($C_u$) and at the station level ($C_s$). As was indicated above, if a contractor performs repair work using the station's equipment, the equipment operation and maintenance costs ($C_E$) are also included in the overhead. Overhead costs are expenses associated with providing conditions for the organization, management, and maintenance of the electricity generation process.

$$C_{oc} = C_u + C_s + C_E \, , \ [rubles] \tag{A3}$$

Rates of overhead costs are defined based on the existing organizational and technical conditions of the station, as a percentage of the wages of operation and maintenance personnel ($C_W$) minus compensation payments.

Non-production costs ($C_{nc}$) are product distribution expenses. They include packaging costs at finished product warehouses, product delivery and loading costs, and other costs related to the process of product distribution. Packaging and shipping costs are included in the production costs of corresponding types of products directly. If such allocation is impossible, they are among between individual types of products based on their weight, volume, or production cost.

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
