# Peer review of "Economic Efficiency of the Implementation of Digital Technologies in Energy Power"

_sustainability, doi:10.3390/su142215382_

Round 1
Reviewer 1 Report
Dear Author(s),
Thank you for the perspective illustrated through your study. Your theme 'Economic efficiency of the implementation of digital technologies in energy power' is a vital area in the subject of global energy transition and climate neutrality drive.
However, there are major concerns that need to be addressed especially toward improving the quality of the study. Authors are encouraged to pay attention to the following comments
1) The language quality is not sufficient. For instance, in the abstract, you wrote 'The directions of the further research...' instead of 'The directions of further research....'
2) While digitization and environmental-related aspects are parts of the main elements of the study, its important authors provide more robust and clear information about the role of digitization in the Russian economy. Endeavor to use relevant and recent studies to convey your argument. See examples here (i)https://doi.org/10.1002/bse.2932 (ii)https://doi.org/10.1007/s11356-021-14056-5
3)The methodology of the study lacks scientific organization. In the current form, a lot of equations were used but without logical arrangement as to guide the readers toward proper understanding. You might need to revise this section significantly by re-arranging the procedure in a more understandable way.
4) The discussion of the result is also poorly structured because it consists of many short/one-sentence paragraphs, thus rendering the text difficult to follow.
Generally, rigorous work needs to be done in order to improve the quality of the manuscript.
Author Response
Dear colleague,
Thank you very much for sharing your point of view and valuable comments on the paper. We have tried to take them all into account. Please find attached the revised copy of the manuscript.
Victoria Galkovskaya & Marria Volos

Reviewer 2 Report
Any corrections for this article

Author Response
Dear colleague,
Thank you very much for sharing your point of view and valuable comments on the paper. We have tried to take them all into account. Please find attached the revised copy of the manuscript.
Regards,
Victoria Galkovskaya & Mariia Volos

Reviewer 3 Report
Comment 1: Abstract. The abstract should give the appropriate background to the study of this manuscript.
Comment 2: Figure 2 can be replaced with an image format to better represent the relevant content in the image.
Comment 3: What is the main innovation of this study? It is recommended that the authors elaborate on this.
Comment 4: sections of Introduction. The background of the COVID-19 pandemic seems too simple. The background of the impact of energy on energy and economy needs to be considered under the pandemic. There has already been a large amount of literatures discussing this topic. There is a need to better elaborate the background of the COVID-19 pandemic. Please consider citing following papers: (i) https://doi.org/10.1016/j.envres.2021.111990; (ii) https://doi.org/10.1016/j.envres.2021.111637; (iii) https://doi.org/10.1016/j.scitotenv.2020.138915.
Comment 5: The image format of Figure 3 does not show the relevant content well, so we suggest the authors to modify this with flowcharts or other image formats.
Comment 6: The conclusion section should give the main findings of this manuscript and related policy recommendations.
Comment 7: The limitations of this study and directions for future research should be reflected in the manuscript.
Author Response

(The authors gave the same response as above.)

Reviewer 4 Report
At the end of the introduction, the authors mention a Chapter. This is a paper or a Chapter?
Also, a guided tour of the article is missing at the end of the Introduction.
The discussion of results is poor and it is not contrasted with the literature available.
The methodology used for computing the costs is standard.
The conclusions are poor.
I do not see enough novelty in the paper that deserves publication.
Author Response

(The authors gave the same response as above.)

Round 2
Reviewer 1 Report
Thank you.
Author Response
Dear referee,
We have tried to address all the your concerns in a proper way.
Regards,
Dr. Galkovskaya, Dr. Volos

Reviewer 3 Report
The authors have incorporated comments from the first round of review. My concerns from my previous review have been addressed. I would recommend the paper to be accepted for publication.
Author Response

(The authors gave the same response as above.)

Reviewer 4 Report
The authors did not reply considering all the points raised by reviewer. Therefore I am against the publication of this paper.
Author Response

(The authors gave the same response as above.)
